# Protective Role of Self-Regulatory Efficacy: A Moderated Mediation Model on the Influence of Impulsivity on Cyberbullying through Moral Disengagement

**DOI:** 10.3390/children10020219

**Published:** 2023-01-26

**Authors:** Marinella Paciello, Giuseppe Corbelli, Ileana Di Pomponio, Luca Cerniglia

**Affiliations:** Faculty of Psychology, Uninettuno University, 00186 Rome, Italy

**Keywords:** adolescence, cyberbullying, impulsivity, moral disengagement, self-efficacy

## Abstract

During online interactions, adolescents are often exposed to deviant opportunities. In this context, the capacity to regulate one’s behavior is essential to prevent cyberbullying. Among adolescents, this online aggressive behavior is a growing phenomenon, and its deleterious effects on teenagers’ mental health are well known. The present work argues the importance of self-regulatory capabilities under deviant peer pressure in preventing cyberbullying. In particular, focusing on two relevant risk factors, i.e., impulsivity and moral disengagement, we examine (1) the mediation role of moral disengagement in the process leading to cyberbullying from impulsivity; (2) the buffering effect of the perceived self-regulatory capability to resist deviant peer pressure in mitigating the effect of these impulsive and social–cognitive dimensions on cyberbullying. Moderated mediation analysis was performed on a sample of 856 adolescents; the results confirm that the perceived self-regulatory capability to resist peer pressure effectively mitigates the indirect effect of impulsivity through moral disengagement on cyberbullying. The practical implications of designing interventions to make adolescents more aware and self-regulated in their online social lives to counter cyberbullying are discussed.

## 1. Introduction

Cyberbullying represents a growing phenomenon in contemporary technological society. It comprises different forms of online aggression (e.g., flaming, exclusion, harassment, and trickery) enacted by individuals or groups through digital media, resulting in harm or discomfort to others [1]. Its negative psychological effects are well known. Indeed, a large body of research has attested that cyberbullying is associated with serious behavioral and mental health problems [2,3,4,5]. 

Among the key factors affecting the likelihood of cyberbullying, the literature in this field has highlighted the pivotal role of moral disengagement [6,7], defined as the set of social–cognitive mechanisms that allow the legitimization and justification of aggressive behaviors [8,9]. Under certain circumstances, moral disengagement mechanisms can “darken” moral purpose and deactivate or diminish moral control. Indeed, these cognitive stratagems allow people to avoid self-blame and protect their moral image by refraining and distorting behavior and its consequences and by externally dislocating the responsibility for their actions. The relationship between moral disengagement and cyberbullying is grounded on the well-established findings on the role of moral disengagement in understanding aggressive behavior during adolescence [10,11,12] and it has already been extensively demonstrated [6,13,14]. By using moral disengagement mechanisms, “otherwise good” youth engage in cyberbullying [15]. One possible reason is the influence exerted by peers on online problematic behaviors [16] and on moral disengagement [17]. Indeed, moral disengagement as individual cognition influenced by social–environmental systems mediates the relationship between peer pressure and cyberbullying [18].

To understand the role of MD in cyberbullying it is also necessary to bear in mind that cyberbullying could be an impulsive form of behavior. As suggested by the general aggression model and its application to bullying [6,19], triggered by situational affordances an adolescent could engage in an aggressive and impulsive reaction against another individual. This seems particularly true in the case of temperamental traits. In the case of the present study, we have considered impulsivity as an individual proneness to act out impulsive and unplanned responses to stimuli without considering their potential negative effects. The consolidated literature has suggested that impulsivity is associated with problematic internet uses, including cyberbullying [20]. In fact, it has been found that impulsivity is linked with less helping behavior in perpetrators and bystanders of cyberbullying and that it is longitudinally correlated with cyberbullying [21]. Additionally, Safaria [22] and Zych [23] found that cyberaggression is associated with impulsivity, and low quality of social relations online and offline. Several studies have also posited that cyberbullying is very frequent in adolescence because youths (especially males) still have to complete the maturation of prefrontal cortex circuits. These cerebral networks are crucial for downplaying adolescents’ impulses to act out. If these circuits are not fully mature, impulsive behaviors and risk-taking are prevalent in youths. Moreover, based on the related constructs, such as cognitive impulsivity, irritability, or/and trait anger, we can expect that impulsivity influences moral disengagement mechanisms that could justify those reactive forms of aggression. This is in line with longitudinal study findings [24] attesting that irritability (as an individual tendency containing impulsivity features) is an antecedent of moral disengagement, and findings showing the influence of negative emotional activation on moral disengagement [25,26,27]. 

Based on these assumptions, we hypothesize that (see Figure 1):

**H1.** 
*Impulsivity is positively associated with participation in cyberbullying behaviors;*


**H2.** 
*Moral disengagement is also positively associated with cyberbullying;*


**H3.** 
*As impulsivity increases, moral disengagement also increases;*


**H4.** 
*Moral disengagement mediates the relationship between impulsivity and participation in cyberbullying behaviors.*


In exploring the influence of impulsivity on cyberbullying through moral disengagement we also want to examine how and if it is possible to prevent this process. Thus, our second aim is to examine a potential individual protective factor intervening as a moderator in this impulsive–cognitive–behavioral chain. 

Specifically, we hypothesize that resistive regulatory self-efficacy beliefs, namely beliefs adolescents hold about their capacity to resist peer pressures towards transgression [28], can be particularly important in contrasting cyberbullying and mitigating the process leading to it. The perceived capability to resist peer pressure could be a crucial dimension to buffer the effect of moral disengagement on cyberbullying because it can capture the individual vulnerability/resistance to being influenced by negative salient peer pressures. It is noticed that in the presence of peers, adolescents are more inclined to take risks to gain social rewards [29] and that peer pressure can foster moral disengagement and cyberbullying [18]. However, according to an agentic perspective of adolescence [30], adolescents are also able to exert a self-influence on one’s way to behave, reflect, and regulate themselves (and learn to do it). This social cognitive dimension captures adolescents’ perceived capabilities to regulate their moral behavior in tempting situations [31]. In particular, self-efficacious individuals are more capable of self-monitoring and self-judging the pattern of behavior, anticipating its possible consequences and the affective reactions to one’s behavior [32]. Thus, we hypothesize that if the perceived self-efficacy in resisting peer pressure of adolescents is high, then the effects of moral disengagement on cyberbullying should be low (Figure 1). The idea is consistent with previous studies attesting to the protective role of this self-efficacy in hindering deviant and risky behaviors [33,34,35]. However, in the majority of studies, with few exceptions (e.g. [36]), self-efficacy in regulating moral conduct under unethical pressures has only been seen as an antecedent and not a moderator. In this present contribution, we have considered the interactive effects of perceived self-efficacy on moral disengagement as well as cyberbullying. Indeed, as a protective factor self-efficacy could intervene just when the risk of engaging in negative behaviors is high. In accordance with this perspective, recent findings have attested that self-regulatory capabilities become crucial as main guides and deterrents in moderating cyberbullying [37]. Individual differences in self-regulatory capacities could hinder the degree to which deviant peer pressure influences different risk behavior [38,39,40]. In particular, poor self-regulatory capacity increases the likelihood of risk participation [41,42,43]. In this regard, Gardner and colleagues [44] have shown how self-regulatory capacities may moderate the association between peer deviance and adolescent antisocial behavior after controlling for previous levels of antisocial problems. In line with previous findings [39,45], this result suggests that adolescents with low self-regulatory capacities are more vulnerable to peer influence, as they tend to rely on external structures to regulate their emotional and behavioral functioning [38]. On the contrary, adolescents high in self-regulatory capacities are more able to resist the deviant temptations and keep track of long-term goals despite contingent rewards from the peer network. Overall, considering self-efficacy as a personal self-evaluative scheme derived from past and present experience, it could be expected that individual differences in perceived capabilities of resisting deviant pressure can inform how adolescents regulate and manage their disengaging processes leading to cyberbullying. For this reason, we hypothesize that (see again Figure 1):

**H5.** 
*Self-regulatory efficacy negatively influences cyberbullying;*


**H6.** 
*Self-regulatory efficacy is also negatively linked to moral disengagement;*


**H7.** 
*Self-regulatory efficacy moderates the relationship between moral disengagement and cyberbullying behaviors: the association between the two becomes weaker as self-efficacy increases.*


Thus, its potential moderating role could be useful for designing interventions aimed to develop individual self-regulation and a meta-reflection on it, because, as highlighted by Bandura, competent functioning requires both capabilities and the beliefs to use them effectively [46]. 

## 2. Materials and Methods

### 2.1. Participants

The present study is part of a research project regarding the use of technology in adolescence. With the aim of recruiting an adequate sample, the willingness of some Italian schools to participate in the research project was demanded, and their students were included as a convenience sample.

The final sample consisted of 853 students (M_age_ = 14.7, SD_age_ = 1.7), 45.9% of whom were female, from two high schools (72.1%) and two middle schools (27.8%). All the investigations were carried out following the rules of the revised Declaration of Helsinki. 

### 2.2. Procedure

Following the approval of the project by the Ethics Committee of the university to which the authors are affiliated (PSICDF_20180201), the project was approved by the head teachers and boards of education of each school involved. The teachers of the schools received an explanatory document detailing the research procedure and measurement scales included in the questionnaire for the students, while the parents (or legal guardians) also received a detailed handout with information regarding procedures related to privacy and the way personal data would be processed, and were requested to explicitly provide informed consent in order for their children to participate in the project. The sessions for administering the questionnaire were therefore scheduled in accordance with the teaching staff, depending on the curricular timetable provided by the respective school. At the appointed time to conduct the research, students were invited to the institutes’ computer labs, where filling out the online questionnaire took them no more than an hour. The entire administration process was closely supervised by at least one researcher along with a classroom teacher. After data collection, the results of the preliminary analyses were shared back with the teachers through a research report and an on-site seminar.

### 2.3. Measures

#### 2.3.1. Impulsivity

The Italian version of the 30-item Barratt Impulsiveness Scale (BIS-11) [47,48] was used to assess the construct by including the three dimensions of attentional/cognitive impulsiveness (eight items), motor impulsiveness (eleven items), and non-planning impulsiveness (another eleven items). The Italian version of BIS-11 has shown an adequate internal consistency (α = 0.79). An example item is “I am a person who does things without thinking”. Participants were asked to rate the frequency of impulsivity-related behaviors by rating 30 items on a scale from 1 (Rarely/Never) to 4 (Almost Always/Always), with an overall total score ranging from 30 to 120. No missing data were found in the final dataset. The Cronbach reliability coefficient for the entire scale on the considered sample evidenced a good level of internal consistency (α = 0.81).

#### 2.3.2. Cyberbullying Perpetration

The perpetration of cyberbullying was assessed using 14 items developed by Palladino, Nocentini, and Menesini [49]; through this instrument, adolescents were to report how often they engage in cyberbullying behaviors using a 5-point Likert scale (ranging from 1 = Never, to 5 = Several times a week). The target behavior is assessed through four areas: three items measure verbal (written) cyberbullying, another three items are for visual bullying, four items instead assess cyberbullying through impersonation, and another three items bullying through exclusion. The original scale has shown a good level of internal consistency (α = 0.89). An example item is “I sent messages to someone I didn’t like, trying to intimidate him/her”.

No missing data were found in the final dataset. For the sample under consideration, Cronbach reliability coefficient for the entire scale evidenced an excellent level of internal consistency (α = 0.90).

#### 2.3.3. Moral Disengagement

Moral disengagement was assessed with the 14-item scale designed by Pozzoli, Gini, and Vieno [50], taking into account eight mechanisms along four loci: six items pertain to the behavioral locus (cognitive restructuring), three items to the agency locus (minimizing one’s agentive role), two items to effects/outcome locus (disregarding/distorting consequences), and another three to victim locus (blaming/dehumanizing the victim). An example item from the original scale, which shows an internal consistency of 0.91, is “to insult annoying people is just to teach them a lesson”. Adolescents were asked to report their level of agreement with each item on a 5-point Likert scale (from 1 = “Not agree at all” to 5 = “Completely agree”). No missing data were found in the final sample. The Cronbach reliability coefficient for the sample under consideration indicated a good level of internal consistency (α = 0.80).

#### 2.3.4. Self-Regulatory Efficacy

Self-regulatory efficacy was evaluated using an 8-item scale [28], each rated on a 5-point Likert scale ranging from 1 (“Not at all capable”) to 5 (“Completely capable”) in order to measure the perceived capacity to rebuff peer pressure to engage in wrong behaviors. The scale has originally shown a good level of internal consistency (α = 0.89). An example item is “How capable do you think you are of evading the insistence of friends who ask you to do things you think would be best avoided?”. Again, no data were missing in the final dataset. In the present study, Cronbach’s α for the scale was 0.80.

### 2.4. Data Analysis

After checking the acceptability of the internal consistency of each scale, in order to verify the posited moderated mediation, first, the significance of the mediating role played by moral disengagement between impulsivity and cyberbullying was tested. Second, the moderation hypothesis was tested and the regression system was subsequently re-specified to include only significant interaction terms; in this way, conditional effects were calculated. Indirect effects were estimated by means of bootstrapping at a 95% confidence interval with 10,000 bootstrap samples [51]. All analyses were conducted using R [52] within the RStudio development environment [53], by means of the dplyr [54], lavaan [55], and psych [56] packages.

## 3. Results

Table 1 shows the descriptive statistics and correlations for the variables included in the proposed model. As both skewness and kurtosis for the distribution of cyberbullying were greater than the ∣2∣ cut-off [57,58], assessment of the significance of the parameters in all steps of the analysis was performed on the basis of 95% confidence intervals estimated using the bias-corrected bootstrap method (with 10,000 samples) to account for violation of the assumption of normality [59]. It can be seen that impulsivity, moral disengagement, and cyberbullying are significantly positively correlated with each other, while each of the three constructs is negatively correlated with self-regulatory efficacy. 

First, the results support the significant and positive association between impulsivity and cyberbullying (H1; β = 0.120 [0.044], 95% BCI = [0.033, 0.206]). Following the introduction of moral disengagement as a mediator, it is shown that when moral disengagement increases, the incidence of cyberbullying behaviors also increases (H2; β = 0.365 [0.056], 95% BCI = [0.255, 0.474]). Similarly, the link between impulsivity and moral disengagement is also significantly positive (H3; β = 0.276 [0.037], 95% BCI = [0.203, 0.349]). The introduction of moral disengagement between impulsivity and cyberbullying leads to non-significance of the direct effect between the focal predictor and the outcome (Table 2), while the indirect effect from impulsivity to cyberbullying through moral disengagement is significant (H4; ind.eff. = 0.101 [0.018], 95% BCI = [0.065, 0.137]). 

Model results (χ^2^ = 1.582, df = 2, p = 0.453; CFI = 1.000; TLI = 1.000; RMSEA = 0.000 (90% CI = 0.000 – 0.063), p = 0.874; SRMR = 0.009) after the introduction of the moderator as a predictor of both endogenous variables show that self-regulatory efficacy is negatively associated with cyberbullying perpetration (H5; β = −0.234 [0.045], 95% BCI = [−0.322, −0.146]), and also that when self-efficacy rises, moral disengagement decreases accordingly (H6; β = −0.346 [0.036], 95% BCI = [−0.415, −0.276]). 

Having included the posited interaction term in the final regression system, regulatory self-efficacy is found to significantly moderate the effect of moral disengagement on cyberbullying (Figure 2), as hypothesized (H7; interaction: β = −0.132 [0.066], 95% BCI = [−0.261, −0.003]). 

Considering indirect conditional effects (Table 3), it can be observed that impulsivity influences cyberbullying behaviors through the action of moral disengagement mechanisms more strongly in those individuals with low self-regulatory efficacy (β = 0.058 [0.018], 95% BCI = [0.022, 0.093]), and to a lesser extent for those with an average self-regulatory efficacy score (β = 0.037 [0.012], 95% BCI = [0.014, 0.061]). However, the same relationship does not hold for those individuals with a high self-regulatory efficacy value (Figure 3), for whom the indirect effect is not significant (β = 0.017 [0.016], 95% BCI = [−0.014, 0.049]). 

Regarding control variables, gender and age were considered as covariates during all steps of the analysis; the results indicate that gender significantly impacts moral disengagement, with males more likely to adopt its mechanisms (β = −0.176 [0.030], 95% BCI = [−0.234, −0.117]); age, on the other hand, is significantly related only to cyberbullying perpetration, with younger people reporting fewer behaviors of this nature (β = −0.091 [0.031], 95% BCI = [−0.152, −0.031]).

## 4. Discussion

This study increases our knowledge about the important role played by personal resources in self-regulative processes in interrupting impulsive engagement in cyberbullying during online peer interactions. The results of our research provide a piece of clear evidence of the non-linearity of the impulsion–moral disengagement–aggression path. In particular, we showed how the direct link between impulsivity and cyberbullying is no longer significant after the introduction of moral disengagement as a mediator, thus providing a possible explanation for the link between impulsivity and cyberbullying through the disinhibitory role of moral disengagement. Although this result was expected, to our knowledge, this is the first study attesting to a mediational role of moral disengagement in the relationship between impulsivity and cyberbullying. Furthermore, after testing the moderating action of perceived self-regulatory efficacy on the possible links leading from impulsivity to cyberbullying even through the mediator, we highlighted how individual differences in the level of self-efficacy only play an influential role on the path between moral disengagement and cyberbullying. In more depth, even when individuals might be high in moral disengagement, our findings suggest that this condition does not necessarily mean that adolescents fall unavoidably into online negative behavior. This result is in line with previous findings on adulthood samples showing that moral disengagement does not automatically increase the likelihood of engagement in unethical behavior when individuals perceive themselves as able to regulate their moral behaviors [36]. In the case of adolescence, those high in self-efficacy in resisting negative peer pressures should be more able to self-reflect on themselves and their conduct and anticipate the negative consequences of cyberbullying. On the contrary, adolescents low in self-efficacy in resisting peer pressure are less aware of the internal (e.g., impulsive reaction) and social forces (e.g., peer pressures) that can increase moral disengagement recourse. However, this does not mean that self-efficacious adolescents are infallible, but they probably have major resources to cope with moral challenges and are less likely to be at the mercy of moral disengagement and impulsive/automatic reactions.

Three fundamental developmental issues are at the basis of this reasoning. The first issue concerns the fact that acting out in adolescence is not necessarily a maladaptive behavior, developmentally speaking. Youths’ maturation and identity structuring are rooted in experimentation, limit challenging, and risky behaviors [60]. Thus, adolescents might engage in negative behaviors under the stimulus of a developmentally appropriate drive, which should be dampened by mature emotion regulation processes. The second issue concerns the fact that in adolescence individuals gradually shift from their parents to their peers as reference figures. Therefore, their impulsive behaviors usually occur among a group of fellow adolescents, who act as role models and attachment figures [61]. The third issue regards the emerging capacities of a more complex and mature competence in mentalizing during adolescence. During adolescence, youths gradually acquire the ability to reflect on their impulses and actions, more broadly considering possible precursors and outcomes of their activities [62].

Concerning practical implications, these results could be helpful because: (1) they inform about the relation among temperamental and social cognitive mechanisms leading to cyberbullying; (2) they highlight the importance of improving self-regulation and meta-cognitive beliefs on the self (i.e., self-efficacy) to prevent the effects of risky individual dimensions (i.e., impulsivity and moral disengagement) underlying cyberbullying. Specifically, on one hand, cyberbullying intervention can aim to increase the awareness on how this kind of online aggressive behavior can result from impulsive internal reactions and cognitive processes often activated by external conditions. On other hand, it is important to integrate existing intervention on emotional regulation with modules on behavioral control increasing self-efficacy. Practically, self-efficacy can be improved by operating directly on its source (mastery experience, vicarious experience, persuasion, improving positive emotions) and its development is crucial for behavioral change [63]. Thus, these results could be used for designing educational and prevention programs in which adolescents have the opportunity to observe themselves and others regarding the complexity of ethical decision-making in online scenarios and to reflect on the capabilities needed to master online challenges. For example, the use of realistic scenarios could help the development of self-regulative capabilities when adolescents are tempted to engage in aggressive online behavior. These simulative scenarios can also offer adolescents the opportunity to discuss their own experiences and to think together about alternatives and a constructive possibility to deal with the temptation to disengage and adopt impulsive negative behavior during online social interactions. The focus of intervention on self-beliefs related to self-regulation should allow adolescents to become more aware of themselves and their personal resources. In sum, cyberbullying and its underlying processes could be prevented by promoting reflection on individual agentic roles in managing internal and external pressure, and on personal power in amplifying or diminishing harmful online dynamics. 

### Limitations and Future Prospects

Notwithstanding the goodness of the present results, our findings need to be tested by using different research designs and methodologies. First, the present study comprised only self-reported measures that can be subject to social desirability biases. Thus, future studies should also consider other kinds of measures, such as physiological indices and observed behaviors. Second, in the present study we have been unable to explore the causal links between impulsivity and regulatory self-efficacy. As suggested by correlational analyses, these two dimensions are significantly associated. However, their relationship needs to be understood over time by adopting a perspective approach and longitudinal design. Third, considering the specificity of online settings and cyberbullying dynamics, other variables should be examined. For instance, the investigation of online peer pressure and technological affordances could provide a clearer picture of how individuals, environment, and behavior can influence each other over time. Finally, future clinical experimental studies should verify the efficacy of intervention aimed to improve self-regulatory skills, considering the individual temperamental traits and the contextual and social factors not included in the present study. 

## 5. Conclusions

Over the years, much theorizing and research have been devoted to designing proper interventions aimed to prevent and contrast problematic behaviors under tempting situations (e.g., [64]). In this regard, particular attention has been given to protective factors related to self-regulation that enable adolescents to properly manage risky situations and to disengage from them [33,41,44]. Engagement in harmful and damaging behavior rarely occurs in a vacuum [38,40]. In the specific case of cyberbullying, online social dynamics can potentially jeopardize the personal control, especially when an adolescent’s impulsivity trait is high. This temperamental characteristic can indeed facilitate the activation of moral disengagement mechanisms that are already influenced by online affordances [15]. However, in accordance with an agentic perspective of human development, this study suggests that self-efficacy can hinder impulsive online aggressive behavior under internal (e.g., impulsive reactions) and external pressures (e.g., peer pressures). Promoting self-reflection on current and potential self-regulative capabilities can make a difference when emotional arousal is saliently activated in the online interpersonal dynamics in which cyberbullying could be socially rewarded.

## Figures and Tables

**Figure 1 children-10-00219-f001:**
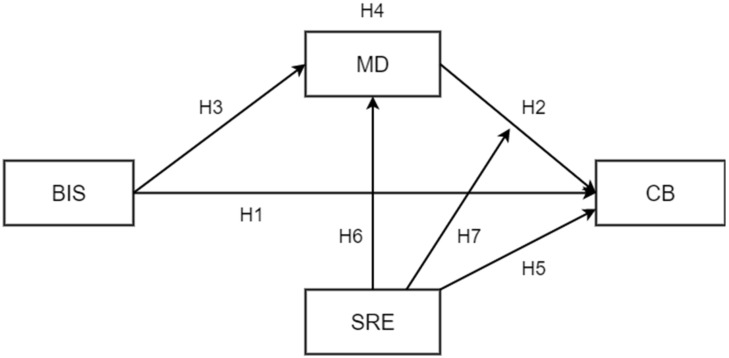
Hypothesized moderated mediation model; the link between impulsivity and cyberbullying is explained by moral disengagement, while the relationship between moral disengagement and cyberbullying is moderated by individual differences in self-regulatory efficacy. Note. BIS = Barratt Impulsiveness Scale; MD = Moral Disengagement; CB = Cyberbullying; SRE = Self-Regulatory Efficacy.

**Figure 2 children-10-00219-f002:**
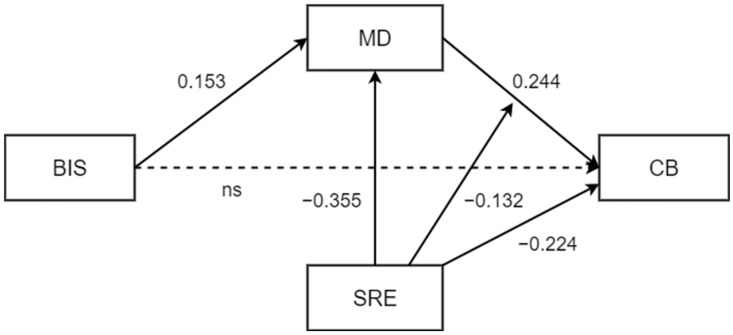
Final moderated mediation model. Note. Standardized path estimates are displayed. The effects of controls (gender and age) are not shown for the sake of clarity. Dashed lines indicate non-significant paths. BIS = Barratt Impulsiveness Scale; MD = Moral Disengagement; CB = Cyberbullying; SRE = Self-Regulatory Efficacy.

**Figure 3 children-10-00219-f003:**
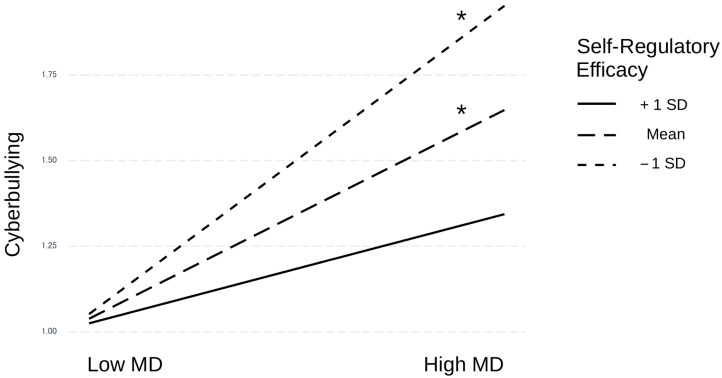
Interaction effect of moral disengagement and self-regulatory efficacy on cyberbullying. Note. * = significant coefficient.

**Table 1 children-10-00219-t001:** Descriptive statistics and zero-order correlations for the relevant variables.

	M	SD	1	2	3	4	5
1. Gender	1.46	0.50	-				
2. Age	17.7	1.70	−0.01	-			
3. BIS	64.10	10.01	−0.02	−0.05	-		
4. MD	1.86	0.55	−0.21 ***	−0.06	0.28 ***	-	
5. CB	1.21	0.39	−0.08 *	0.07*	0.12 ***	0.37 ***	-
6. SRE	3.89	0.79	0.11 **	0.05	−0.37 ***	−0.42 ***	−0.33 ***

*Note.* BIS = Barratt Impulsiveness Scale; MD = Moral Disengagement; CB = Cyberbullying; SRE = Self-Regulatory Efficacy. * *p* < 0.05, ** *p* < 0.01, *** *p* < 0.001.

**Table 2 children-10-00219-t002:** Total and indirect effect (without the moderator).

	Effect									
Path	Total			Direct			Indirect			% tot
	β	SE	95% BCI (lower, upper)	β	SE	95% BCI (lower, upper)	β	SE	95% BCI (lower, upper)	
bis→bul	0.120	0.044	0.033, 0.206 *							100
bis→md→bul				0.020	0.047	−0.073, 0.113	0.101	0.018	0.065, 0.137 *	84

Number of bootstrap resamples: 10,000; * = the effect is significant.

**Table 3 children-10-00219-t003:** Conditional effects.

	Indirect effect								
Path	−1SD			Mean			+1SD		
	β	SE	95% BCI (lower, upper)	β	SE	95% BCI (lower, upper)	β	SE	95% BCI (lower, upper)
bis→md→bul	0.058	0.018	0.022, 0.093 *	0.037	0.012	0.014, 0.061 *	0.017	0.016	−0.014, 0.049

Number of bootstrap resamples: 10,000; * = the effect is significant.

## Data Availability

The dataset supporting the results of this study is available upon motivated request to the corresponding author (G.C.). The data are not publicly available because, although completely anonymous, their public disclosure is not specifically mentioned in the informed consent provided by the participants on the use of confidential data.

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
