# Peer review of "Protective Role of Self-Regulatory Efficacy: A Moderated Mediation Model on the Influence of Impulsivity on Cyberbullying through Moral Disengagement"

_children, 2023, doi:10.3390/children10020219_

Round 1

Reviewer 1 Report

The manuscript is simple but brings many study variables and the relationships are interesting.

The protective role of regulatory self-efficacy: a moderated mediation model on the influence of impulsivity on cyberbullying through moral disengagement (children-2164678)

Initial comment: In general, the manuscript seems to me an appropiate paper to be published. The authors made a sufficiently consistent introduction and the analysis methods are adequate. The discussion responded to a greater or lesser extent to the research problems.

However, I would like to make some clarifications that would improve the quality of the article.

Please answer each of the sections separately:

Abstract: Add some more introduction and research problem in the abstract. At the same time, it is better to sort the keywords alphabetically.

1. Introduction: The introduction is appropriate and summarizes the objective of the study. However, the hypotheses or research problems must be established at the end of the section. In addition, figure one should be clearer in line with Hayes (2018) establishing the directions of the proposed model.

2. Material and methods:

Participants: It must be indicated how the sample was selected for the investigation. Informed consent must be mentioned as same as the code or ethic guidliness.

2.1 Participants and Procedure: The sections must be differentiated and not together. The "Procedure" section should be expanded a little more. A detailed description of the questionnaires as well as original Cronbach's Alpha should be mentioned.

Statistical Analysis: Statistical analysis should be written more clearly relating it to the research objectives or intended results. Its better to show 10000 bootstrapping mode.

3. Results: The statistical treatment is simple but adequate with the aims and scope of the manuscript.

4. Discussion: The discussion is correct according to the information provided, although I still think that a better statistical treatment would considerably improve the manuscript.

5. Conclusions: Appropriate conclusions in line with the results and the discussion. The sections "Limitations of the study" and "Future prospects" must be included.

Minor revisions:

a)      Sort the keywords alphabetically

Final comment: I would like to strongly encourage authors to reformulate their manuscript with the changes made in this document.

Thank you very much.

Author Response

Dear Reviewer,
Thank you for taking the time to review our manuscript ("The protective role of self-regulatory efficacy: a moderated mediation model on the influence of impulsivity on cyberbullying through moral disengagement"; Manuscript ID: children-2164678).  
We greatly appreciated your careful consideration and valuable feedback.

Abstract: Add some more introduction and research problem in the abstract. At the same time, it is better to sort the keywords alphabetically.

In response to your suggestion, we have provided some context lines in the abstract to better explain the research problem, highlighting the critical importance of cyberbullying in digital contexts. Additionally, as requested, we have listed the keywords in alphabetical order.

1. Introduction: The introduction is appropriate and summarizes the objective of the study. However, the hypotheses or research problems must be established at the end of the section. In addition, figure one should be clearer in line with Hayes (2018) establishing the directions of the proposed model.

Thanks to your request we were able to clarify the description of our study. The hypotheses are now presented accurately in accordance with figures and results (see line 74; line 134; Figure 1 and 2).

2. Material and methods:
Participants: It must be indicated how the sample was selected for the investigation. Informed consent must be mentioned as same as the code or ethic guidliness.
2.1 Participants and Procedure: The sections must be differentiated and not together. The "Procedure" section should be expanded a little more. A detailed description of the questionnaires as well as original Cronbach's Alpha should be mentioned.

As requested, we have separated the two paragraphs; around line 162 you can find the point where we make explicit the importance of informed consent asked of the parents of participants. We have expanded the description of the questionnaires by including an example item, as well as the assessment of internal consistency provided by the authors of the questionnaires themselves.

Statistical Analysis: Statistical analysis should be written more clearly relating it to the research objectives or intended results. Its better to show 10000 bootstrapping mode.

As mentioned in response to the first comment, we specified the statistical steps followed so as to clarify remaining ambiguities, tying the analyses to the research objectives. All estimates were calculated, as requested, with 10,000 bootstrap samples.

3. Results: The statistical treatment is simple but adequate with the aims and scope of the manuscript.

Thanks to your requests, we were also able to expand on the description of the statistical treatment in the results paragraph.

4. Discussion: The discussion is correct according to the information provided, although I still think that a better statistical treatment would considerably improve the manuscript.

See above.

5. Conclusions: Appropriate conclusions in line with the results and the discussion. The sections "Limitations of the study" and "Future prospects" must be included.

Finally, we added a paragraph including the limitations of the present study, and future prospects. 
Once again, we would like to express our appreciation for giving us the opportunity to improve our work through your comments.

Sincerely,
Giuseppe Corbelli

Reviewer 2 Report

The researchers worked hard to develop a study focused on the protective role of regulatory self-efficacy: a moderated mediation model on the influence of impulsivity on cyberbullying through moral disengagement. Recruitment for such a study is to be commended. This is important work and adds to the literature focused on how regulatory self-efficacy influence on impulsivity on cyberbullying through moral disengagement

Introduction & Literature Review

·         The introduction describes research on the influence of impulsivity on cyberbullying through moral disengagement.

·         The Introduction section covers the background of the study. The rationale of the study is also explained clearly in the study.

·         In the literature review chapter, previous research is well supported. The literature review is consistent, all-inclusive and critical arguments are well written.

Method

·         Methodology is also described well in the study.

Results & Discussion

         The discussion section lacks a thorough discussion of the findings. The authors did a nice job of putting results in context but needs to address more how their research extends the literature. Add a paragraph or two that addresses what meaning is made from the findings cumulatively.

         More clearly address the limitations of how the samples are not representative of the country.

Conclusion

         Conclusion should be precise.

Author Response

Dear Reviewer,
Thank you for taking the time to review our manuscript ("The protective role of self-regulatory efficacy: a moderated mediation model on the influence of impulsivity on cyberbullying through moral disengagement"; Manuscript ID: children-2164678).  
We greatly appreciated your kind words and your careful consideration and feedback.

Results & Discussion
•         The discussion section lacks a thorough discussion of the findings. The authors did a nice job of putting results in context but needs to address more how their research extends the literature. Add a paragraph or two that addresses what meaning is made from the findings cumulatively.

As requested, we proceeded to further discuss the findings, particularly linking the main results to practical implications.

•         More clearly address the limitations of how the samples are not representative of the country.

In response to your request, we have added a paragraph that includes a thorough examination of the limitations of the present study.

Conclusion
•         Conclusion should be precise.

Thanks to your request we were able to clarify and specify the conclusions, rewording parts of the paragraph to make the association with the research objectives more straightforward. 

Again, we would like to thank you for your appreciation of our research and your kind suggestions for improving our work.

Sincerely,
Giuseppe Corbelli